

# Seafloor geomorphology of western Antarctic Peninsula bays: a signature of ice flow behaviour

Yuribia P. Munoz[1], Julia S. Wellner[1]

[1]Department of Earth and Atmospheric Sciences, University of Houston, Houston, Texas 77204 USA

*Correspondence to*: Y.P. Munoz (ypmunoz@uh.edu)

**Abstract.** Glacial geomorphology has been used in Antarctica to reconstruct ice advance and retreat across the continental shelf during the Last Glacial Maximum. Analogous geomorphic assemblages are found in glaciated fjords and can be used in a similar manner to interpret the glacial history in those areas. In addition, understanding the distribution of submarine

landforms in bays and the local controls exerted on ice flow can help improve ice-sheet models through these complicated drainage areas. We present multibeam swath bathymetry from several bays in the South Shetland Islands and the western Antarctic Peninsula. The submarine landforms are described and interpreted in detail. A schematic model was developed showing the features found in the bays; from flutings and moraines in the inner bay, to grounding zone wedges and drumlins in the middle bay, and streamlined features and meltwater channels in the outer bay areas. In addition, we analysed local

variables in the bays and observe that: 1) bay length and width exert a control on the number of landforms found in the bays, the geometry of the bays dictates the types of features that form; 2) bays with shallower water depths tend to form geomorphic features that suggest rapid decoupling of grounded ice from the seafloor; 3) the smaller seafloor features are present only in the smaller glacial systems indicating that short-lived atmospheric and oceanographic fluctuations, responsible for the formation of these landforms, are only recorded in these smaller systems; and 4) meltwater channels are

abundant on the seafloor, however some are subglacial, carved in bedrock, and some are modern erosional features, carved on soft sediment. Lastly, based on geomorphological evidence, we propose the features found in some of the inner bay areas were formed during a recent glacial advance, likely the Little Ice Age.

## 1 Introduction

While warming temperatures in the Antarctic Peninsula (AP) have resulted in the retreat of 90% of the regional glaciers (Cook et al., 2014) and the collapse of ice shelves (Morris and Vaughan, 2003; Cook and Vaughan, 2010), recent studies have shown that this region is currently experiencing a cooling trend since the late 1990s (Turner et al., 2016). The AP is a dynamic region that serves as a natural laboratory to study ice flow and the resulting sediment deposits. As the ice flows, it leaves behind glacial geomorphic features on the seafloor; these submarine landforms have been mapped in other glaciated



environments in Antarctica (Anderson et al., 2001; Wellner et al., 2001; Evans et al., 2004; Heroy and Anderson, 2005; Wellner et al., 2006; Livingstone et al., 2013; Hodgson et al., 2014), southern Chile (Dowdeswell and Vasquez, 2013), North America (Dowdeswell et al., 2016) and northern Europe (Ottesen et al., 2005; Ottesen and Dowdeswell, 2006; Ottesen and Dowdeswell, 2009; Dowdeswell et al., 2010) giving insight into the glacial history of each region. Several seafloor features

have been mapped west of the AP in the continental slope and continental shelf (Dowdeswell et al., 2004; Gales et al., 2013), the South Shetland Islands (Milliken et al., 2009; Simms et al., 2011), South Georgia Island (Hodgson et al., 2014), Bransfield Strait (Canals et al., 2000; Canals et al., 2002), Gerlache Strait (Evans et al., 2004), south of Anvers Island (Domack et al., 2006) and Marguerite Bay (Ó Cofaigh et al., 2002; Anderson and Fretwell, 2008; Livingstone et al., 2013). However, the seafloor geomorphology in western AP bays has not been described in detail, except for a few locations

(Garcia et al., 2016; Munoz and Wellner, 2016; Wolfl et al., 2016). Although most of the data we present is publically available, this is the first instance, to our knowledge, where a detailed description of the seafloor geomorphology of a large number of western AP fjords has been completed.

We present multibeam swath bathymetry data collected in several bays in the western Antarctic Peninsula, the South

Shetland Islands, and Anvers Island (Fig. 1). We map the glacial geomorphic features on the seafloor of various bays in order to understand the controls of ice flow dynamics in each location. Although we do not provide any age constraints, submarine landforms are used to interpret relative timing of deglaciation in the bays.

## 2 Regional Setting

### 2.1 King George Island

The northernmost study area is Maxwell Bay, located in King George Island, the largest island of the South Shetland Islands (SSI) archipelago. The SSI are separated from the AP by the Bransfield Strait (Fig. 1). The region has a temperate to sub-polar glacial setting (Yoon et al., 2004). King George Island experiences a maritime climate, with little changes in air temperature throughout the year (average of -1.8° C, minimum of -5.7° C in July, and maximum of 2° C in January), mean annual precipitation of 1249 mm, with high summer rain and high relative humidity (average 88.7%) (Lee et al., 2008;

KOPRI, 2014; Moon et al., 2015; Fernandez et al., 2016). King George Island is covered by an ice cap (approximately 150-200 m thick), and Maxwell Bay has rocky to gravelly beaches (Griffith and Anderson, 1989; Simms et al., 2011). Sediment accumulation rates vary throughout Maxwell Bay: 5.5 mm yr$^{-1}$ in inner Collins Harbor, 5.2-6.6 mm yr$^{-1}$ in inner Marian Cove, and 1.6 mm yr$^{-1}$ in outer Potter Cove (Boldt et al., 2013). The rock outcrops in the island include upper Jurassic volcanic rocks, lower Tertiary to upper Cretaceous Andean intrusive rocks (including adamellite, diorite, granite, gabbro,

granodiorite, quartz diorite, and tonalite), upper Tertiary volcanic rocks, Pliocene conglomerates and Quaternary volcanic rocks (Adie, 1969).



## 2.2 Trinity Peninsula

The northernmost tip of the AP is known as the Trinity Peninsula (Fig. 1). Pereira et al. (2013) classified the climate of Hope Bay (located on the northern tip of the AP) as cold, dry semi-polar. Annual air temperature at Esperanza Research Station (located in front of the Antarctic Sound) have been measured to range between -30.6° C and 11.8° C, with an average of -5.1° C, and an annual precipitation of 250 mm between 1952 and 2010 (Pereira et al., 2013; Schaefer et al., 2016). Strong winds, of up to 380 km hr$^{-1}$, have been recorded at Esperanza Research Station (Antarctic Treaty Consultative Meeting, 2015). Boldt et al. (2013) measured a sediment accumulation rate of 3 mm yr$^{-1}$ Hope Bay. Outcrops in the northern AP are mainly Carboniferous sedimentary rocks and lower Tertiary to upper Cretaceous Andean intrusive rocks (diorite, granite, and gabbro) (Adie, 1969).

## 2.3 Anvers Island

Anvers Island is part of the Palmer Archipelago, located west of the Danco and Graham Coasts, separated from the AP by the Gerlache Strait (Fig. 1). Air temperatures in the austral summer reach up to 6-7.5° C, while in the winter temperatures average -5° C in Anvers Island (Ashley and Smith, 2000). Very high amounts of precipitation characterize the Palmer Archipelago, approximately 1200 mm annually (Griffith and Anderson, 1989; Ashley and Smith, 2000), and up to 2000 mm yr$^{-1}$ in one of the bays, Lapeyrère Bay (Fernandez et al., 2016). Sediment accumulation rates have been measured in Lapeyrère Bay to vary between 2.2-3.2 mm yr$^{-1}$ (Boldt et al., 2013). The outcrops in Anvers Island are mainly lower Tertiary to upper Cretaceous Andean intrusive rocks (granite, diorite, tonallite, gabbro) with some upper Tertiary and upper Jurassic volcanic rocks (Adie, 1969).

## 2.4 Central Antarctic Peninsula

Most of the bays in this study are located in the central AP, in the regions known as the Davis Coast (Hughes Bay and Charlotte Bay), Danco Coast (Andvord Bay and Flandres Bay), and Graham Coast (Collins Bay and Beascochea Bay) (Fig. 1 and figures in supplementary material). The climate in this area is dry, subpolar (Griffith and Anderson, 1989; Boldt et al., 2013). Annual air temperatures vary between slightly above 0° C in the summers to -8° to -11° C in the winters (King et al., 2003). Sea ice covers the bays seasonally, but most areas are sea-ice free during the austral summers (Domack and Ishman, 1993). This region is also characterized by low average annual precipitation because it is located in the rain shadow of the AP plateau (Griffith and Anderson, 1989). However, Fernandez et al. (2016) reported a mean annual precipitation of up to 2900 mm yr$^{-1}$ in Andvord Bay. Sediment accumulation rates vary throughout bays in the western AP: from 2.8 mm yr$^{-1}$ in Charlotte Bay and Flandres Bay, 5.6 mm yr$^{-1}$ in Andvord Bay, to 2.2-7 mm yr$^{-1}$ in Beascochea Bay (Boldt et al., 2013). Rock outcrops in the central AP are mainly lower Tertiary to upper Cretaceous Andean intrusive rocks (gabbro, granodiorite, granite, diorite, and tonallite) and a few upper Jurassic volcanic rocks and Carboniferous sedimentary rocks (Adie, 1969).





## 3 Methods

Multibeam swath bathymetry data were collected on multiple research expeditions to the western AP on the RV/IB
*Nathaniel B. Palmer* (NBP0201, NBP0502, NBP0602A, NBP0703, NBP1001, and NBP1203) and the RV/IB *Araon*
(AR1304). Multibeam soundings were collected in a swath perpendicular to the ship track using a hull-mounted swath
profiler, Simrad EM120 12 kHz with 191 beams for NBP cruises and Simrad EM122 12 kHz with 432 beams for the AR
cruise. The survey data were corrected for anomalous readings and edited to create relief maps. These data sets were merged
using Caris HIPS & SIPS and relief maps were created in ArcGIS. Sets of gridding were completed per bay, 25x25 m, 10x10
m, and up to 5x5 m, depending on the data available. Here we show the best resolution of the data, which in most cases is
25x25 m. Vertical exaggeration was increased between 2x and 5x to aid visualization and identification of geomorphic
features. This compilation of bathymetry in addition to other data sets have recently been published in Boldt et al. (2013) and
Lavoie et al. (2015). In addition to mapping the submarine landforms, we compare them to the local variables of each bay
including latitude, area, length, width, glacier catchment area, and the seafloor lithology, to understand controls on ice flow
behaviour. High resolution shallow subbotom profiles, CHIRP, are also shown. These were collected during NBP0703
throughout the study area. The CHIRP data are used to identify seafloor lithology (sediment type or bedrock) and the
thickness of the sediments overlying the seafloor. The data were collected using a hull-mounted Knudsen 320B/R and it has
been interpreted using SMT Kingdom software.

## 4 Bathymetry Results

We describe seafloor landforms found in the bays of the western Antarctic Peninsula. Figure 2 shows some of these
landform features mapped in the seafloor and Fig. 3 shows the CHIRP facies interpreted. Below we describe the seafloor
geomorphology in four bays in the western AP; maps and cross sections of the other bays in this study can be found in the
supplementary material. Data tables in the supplementary material show the latitude, longitude, bay length and width, bay
area, number of glaciers in each bay, total glacier catchment area, and the submarine landforms found in each bay.

### 4.1 King George Island- Maxwell Bay

Maxwell Bay (62°13.7'S, 58°50.9'W) (Fig. 4) is located in the western end of King George Island. Maxwell Bay is about 15
km long and between 6-15 km wide, and has an approximate area of 140 km$^2$. Maxwell Bay has several embayments; Edgell
Bay, Ardley Harbor, Collins Harbor, Potter Cove (Fig. 4b) and Marian Cove (Fig. 4c). Water depths vary widely from 35 m
in the inner bay to 500 m in the outer bay. The outer bay is U-shaped, with tens of meters of sediment cover (Milliken et al.,
2009; Fernandez et al., 2015). The glacier catchment area around Maxwell Bay is about 92 km$^2$, formed by four glaciers.

The seafloor in northern Maxwell Bay, near Collins Harbor and Ardley Harbor, is a large, shallow platform with water
depths up to 280 m. Water depths increase rapidly to >400 m in the middle of the bay, forming gullies and channels that cut



into the seafloor in this area of Maxwell Bay. Long channels, presumably active, are observed from Edgell Bay and from Marian Cove flowing towards the middle of Maxwell Bay. Large promontories are located between King George Island and Nelson Island and are possibly an area of previous glacial grounding. A few elongated, streamlined hills, parallel to the axis of the bay, are present in the middle of the bay. These elongated features were likely shaped by flowing ice. Sediment thickness in the outer bay are in excess of 100 m (Milliken et al., 2009), and it is assumed that other features carved by flowing ice, if any, are buried. Simms et al. (2011) showed a large sediment fan at the mouth of Maxwell Bay, draining out of the bay into the Bransfield Strait. The fan has a sediment thickness of up to 1000 m and it is located in water depths between 400 and 1400 m.

### 4.1.1 Potter Cove

Potter Cove (62°13.9'S, 58°41.2'W) (Fig. 4b) is an elongated bay in the southeastern edge of Maxwell Bay. Potter Cove is approximately 4 km long and between 1 km wide in the bay head and 2.5 km wide in the bay mouth. Potter Cove has a small area compared to Maxwell Bay, only approximately 7 $km^2$, and with water depths ranging between 25-150 m. Fourcade Glacier drains directly into this bay, however most of it terminates on land. The glacier catchment area is about 20 $km^2$. Ice front retreat of Fourcade Glacier has been approximately 1 km in Potter Cove between 1956 and 2008 (Wolfl et al., 2016), with a greater retreat of grounded ice on the tidewater glacier and much less on the grounded ice on land (Ruckamp et al., 2011).

Potter Cove is separated from Maxwell Bay by a shallow sill, that measures approximately 30 m across. The seafloor geomorphology in Potter Cove is characterized by numerous transverse moraines. Although the multibeam survey covers a small area of the bay, transverse ridges across the bay are abundant in the data set. We have classified the transverse ridges into two sets: 1) large, continuous ridges across the width of the bay, with water depths ranging between 30-40 m, and 2) small, semi-continuous ridges, semi-transverse to the cove, located between the larger sills, with water depths varying between 40-50 m. We interpret the first set of ridges as transverse moraines and the second set of features as crevasse squeeze ridges. The ridges closer to the head of the bay are not straight across the width of the bay, instead the ridges are arcuate and closer together, compared to transverse ridges in the mouth of the bay.

### 4.1.2 Marian Cove

Marian Cove (62°12.8'S, 58°46.1'W) (Fig. 4c) is an elongated bay, in the northeastern edge of Maxwell Bay, north of Potter Cove. The bay is approximately 4 km long and between 1-1.5 km wide, with an approximate area of 5 $km^2$. A single tidewater glacier (with a catchment area of about 15 $km^2$) drains directly into the bay. It is worth noting that this glacier retreated about 1.7 km between 1956 and 2013 (Lee et al., 2008; Moon et al., 2015). Large meltwater and sediment influx into the bay occur in the summer months (Moon et al., 2015).





Transverse ridges across Marian Cove characterize the seafloor topography. Three major transverse ridges divide the bay into a proximal, middle, and outer basin. The proximal basin has the highest water depths, up to 135 m compared to 120 m and 110 in the middle and outer basin, respectively. The outer, most distal transverse ridge separates Marian Cove from Maxwell Bay. Although this feature is found across the width of the bay, depth varies along the ridge from 40 m in the north to 70 m in the south. The middle ridge appears breached, with a possible slope failure deposit located on the west side of the ridge. Unfortunately, the resolution of the data is not clear enough to resolve this feature. However, the deposits have a fan shape and the water depths are shallower in this area. The inner, most proximal ridge is wider than the other two and it could possibly be an amalgamation of more than one ridge. The width of this ridge is approximately 250 m, while the others are approximately 50 m across. The data show hints of other, smaller ridges across the bay, located between the larger ridges, but these are not clear due to the resolution of the data. The inner (most proximal) basin seafloor shows elongated, narrow hills, parallel to the bay axis, unevenly-spaced, and of varying lengths, which are interpreted as flutings. Previous to the late 1950s, this proximal area was covered by ice (Cook et al., 2014; Moon et al., 2015). However, it is unclear whether the ice cover was grounded ice or permanent sea ice. Lastly, a topographic high, about 250 m long, is located on the eastern end of the surveyed area, close to the modern ice front. Water depths along this feature vary between 75-85 m.

## 4.2 Trinity Peninsula- Hope Bay

Hope Bay (63°24.4'S, 57°2.8'W) (Fig. 5) is located in the northernmost tip of the Antarctic Peninsula, in an area known as Trinity Peninsula, and drains out into the Antarctic Sound. The bay is 6 km long, and between 800 m wide in the bay head and 3 km wide in the bay mouth; the bay area is about 11.5 km$^2$. Water depths in Hope Bay vary between 50-320 m. Two large glaciers drain directly into the bay; Depot Glacier (catchment area of 7 km$^2$) and Arena Glacier (catchment area of 16 km$^2$). In addition, other three small, unnamed glaciers (with an average area of 3 km$^2$) also drain into Hope Bay.

The seafloor in inner Hope Bay is characterized by several transverse ridges (Fig. 5b), while the outer bay is characterized by a large, deep basin. Three sets of transverse ridges are present in the inner bay, water depths along the ridges vary between 80-100 m. Each one of these sets of ridges appears as a composite feature, more than one ridge stacked on or near each other. These have been interpreted as terminal moraines, formed by smaller ice re-advances during an overall ice retreat. The seafloor between the innermost moraine and the modern ice front is covered by a series of arcuate-shaped ridges, approximately 1-3m high, which we have interpreted as crevasse squeeze ridges. Previously to the late 1950s, this proximal area was covered by ice (Cook et al., 2014). However, it is unclear whether the ice cover was grounded ice or permanent sea ice. Two large promontories separate the inner bay from the outer bay; in addition, a deep channel flows from the inner bay towards the outer bay basin. This basin is flat topped with water depths averaging 320 m. The outer bay is separated from the Antarctic Sound by a bathymetric high, likely part of a grounding zone that has now been breached. A channel, which presumably formed by meltwater when the ice front was located nearby however still active today, is located in the outer bay, cutting through the bedrock and flowing into the Antarctic Sound.



### 4.3 Anvers Island- Lapeyrère Bay

Lapeyrère Bay (64°25.3'S, 63°17'W) (Fig. 6) is located in northeastern Anvers Island. Lapeyrère Bay is a narrow, elongated bay with water depths varying from 250-740 m, and shallowing towards the fjord walls. The bay is 11 km long, 2 km wide in the bay head and 3.5 km in the bay mouth, with an overall bay area of 32 km$^2$. One large glacier, Iliad Glacier (catchment area of 234 km$^2$), drains into the bay, in addition to other smaller glaciers around the perimeter of the bay, each with an average catchment area of 6 km$^2$.

The seafloor in front of Iliad Glacier is characterized by crude elongated features in the southern area, rugged landscape in the northern area, and meltwater channels throughout (Fig. 6b). The elongated features are asymmetrical and vary in height (4-10 m) and length (200-700 m). The elongated features close to the ice front have an uneven surface, while the features away from the ice front have a gentler relief. Meltwater channels cut through some of these features, and flow from the ice front margin towards the middle of the inner Lapeyrère Bay. Channel widths are up to 200 m and depths up to 30 m. The inner bay is separated from the middle bay by a transverse ridge, interpreted here as a grounding zone wedge, 2.5 km from the Iliad Glacier front. Water depths along the ridge are 280-306 m. The ridge is symmetrical, with gentle slopes on the proximal and distal side. A large meltwater channel, most likely still active, emerges from the grounding zone wedge and flows into the middle bay. The channel is linear, with steep walls and a flat base; it is 250 m across and 30 m deep in the proximal area, and bifurcates towards the middle of Lapeyrère Bay. A streamlined ridge is present in the middle bay, parallel to the bay axis. The ridge is about 2.5 km long, with gentle, lateral slopes, and water depths 340-570 m along the ridge. The seafloor in the middle and outer bay is flat, depths up to 740 m, and gently dipping towards the outer bay. Several slope failures are observed on the steep walls of the fjord.

A small, unnamed, embayment (5 km long, 2 km wide) is connected to Lapeyrère Bay in the northwest. Water depths in this area are 205-360 m, with steep side walls. An unnamed glacier, with a catchment area of 57 km$^2$, drains into this unnamed embayment. A sinuous ridge is present in the embayment mouth, possibly a grounding zone wedge that has now been breached by sediment gravity flows.

### 4.4 Central Antarctic Peninsula- Beascochea Bay

Beascochea Bay (65°31'S, 63°52.2'W) (Fig. 7) is the southernmost bay presented in this study. It is an elongated bay with several embayments in the bay head, three of them are described below. Each one of the described embayments has a large glacier draining directly into it. None of the coves are named and therefore, for the purposes of this paper, we use the name of the glacier to identify the cove; Lever Glacier Cove, Funk Glacier Cove, and Cadman Glacier Cove. Beascochea Bay is approximately 24 km long, 6-13 km wide, with an approximate bay area of 235 km$^2$. Several glaciers drain into this bay along its perimeter; their individual catchment area varies between 1-28 km$^2$.



The inner bay area (Fig. 7c), at the convergence of Cadman and Funk glaciers, is separated from the middle bay by an elongated mount, transverse to the bay length. The depths along this feature vary between 270-370 m, and shallowing towards the fjord walls. This elongated feature in likely a grounding zone wedge. Two major submarine landforms are

present in the seafloor of inner Beascochea Bay, drumlins and flutings, in an otherwise flat basin. The drumlins are tear-drop shaped, length and width varies between 350-1200 m and 120-400 m, respectively. The drumlins have a steep end that points towards Cadman Glacier and a gentler end that points towards the bay mount in the inner bay. The steep ends of the drumlins are between 10-40 m high. The CHIRP shows these features are covered by a thin sediment layer. The drumlins are immediately followed by flutings, located at the gentler end of the drumlins. These flutings are very subtle features on the

seafloor, 8-10 m high and 500-1000 m long. Another set of flutings occurs on the sides of the inner bay basin, along the bay walls. These flutings are parallel to the drumlins and to the first set of flutings. However, the flutings on the basin walls are much shorter in average, 300-400 m long. We hypothesize the differences in size and length are explained by differences in the material forming the elongated feature (till vs. bedrock), or the amount of deformable material available in the area (channel vs. bay walls). Another reason may be the strength exerted by the ice molding these sediments; the bottom of the

glacier would easily mold sediments or till for a longer period of time, while the glacier margins would not exert the same force along the bay walls and therefore the resulting features would be shorter.

The middle of Beascochea Bay is characterized by a rugged seafloor and only two different features are distinguished, channels and a large basin. The channels vary in depth (15-60 m), width (80-200 m), length (200-2000 m), and overall, the

channels are linear, cutting through bedrock. Most of the channels flow directly into a large (surface area 2.5 km$^2$), flat-topped basin, where the average water depth is 670 m.

The bathymetry in outer Beascochea Bay is also rugged, however this region is characterized by an anastomosing network of meltwater channels cutting through bedrock and flowing between streamlined hills, with a few, small, deep, flat-topped

basins located between the elongated features. The surface area of the basins is less than 0.8 km$^2$, and water depths vary between 600-800 m. Numerous meltwater channels are present in the outer bay; they vary in depth (20-50 m), width (140-250 m), and length (100-3000 m). The streamlined hills have an oval shape and they are formed in bedrock. However, these hills are asymmetric and vary in length (0.6-2.5 km) and width (100-800 m). The streamlined features are more elongated closer to the bay mouth and more asymmetric and stubbier away from the bay, out in the open ocean region.

### 4.4.1 Lever Glacier Cove

(65°30.7'S, 63°43.4'W) (Fig. 7b) An elongated bay, 6 km long and 3 km wide, with a bay area of about 16 km$^2$. The largest glacier draining into this cove is Lever Glacier (catchment area of 177 km$^2$), other glaciers draining into the bay are much



smaller (average area is 4 km$^2$) and therefore we assume Lester Glacier had the greatest influence shaping the seafloor in this cove.

A grounding zone wedge separates the cove from the middle of Beascochea Bay. The grounding zone wedge has been
partially surveyed, however it likely connects to Bachstrom Point and into another cove in the bay head (which has not been surveyed). The grounding zone wedge is a ridge, transverse to the cove length axis and has a (surveyed) length of 5 km, but it could be as long as 6 km, accounting for the area not surveyed. Water depths along the ridge vary from 130-180 m, but can reach down to 230 m in breached areas in the ridge. Another grounding zone wedge is present in the cove, proximal to the modern ice front. It is located less than 1 km away from Lever Glacier, and therefore we assume this grounding zone wedge
is relatively modern. It is sinuous with a prominent knob in the middle of the cove. This knob coincides with the deepest area in the cove, enhancing it further in the bathymetry. The grounding zone wedge was only surveyed in the northern area of the cove but it is likely present across the bay. Flutings cover the seafloor along the cove, from the proximal to distal grounding zone wedges. The flutings are not parallel, they are asymmetric, and of varying lengths (400-1400 m) and heights (5-10 m), as well as distance between the fluting crests (90-260 m). Although water depths within the cove vary from 120-320 m, the
flutings are present in the seafloor regardless of water depth. In the CHIRP dataset, the flutings are characterized by a strong surface with no internal reflectors, which we interpret as till (Fig. 3).

### 4.4.2 Funk Glacier Cove

(65°34.8'S, 63°45.4'W) (Fig. 7c) An elongated bay, 4 km long and 2 km wide, with a bay area of about 8 km$^2$. A large
glacier drains directly into this cove, Funk Glacier, with a surface area of 158 km$^2$. Another small glacier, with a surface area of 3 km$^2$, also flows into the cove. Due to differences in size, Funk Glacier largely exerts the greatest influence shaping the seafloor in this cove. Previously to the late 1960s, this cove was covered by ice (Cook et al., 2014). Although, it is unclear whether the ice cover was grounded ice or permanent sea ice, the fact that sea ice covered the bay again in the late 1990s supports the conclusion that fast sea ice covered the cove before the late 1960s, and not grounded ice.

A large grounding zone wedge separates the cove from the main Beascochea Bay, water depths along the grounding zone wedge vary from 136-230 m and down to 270 m in an area where the grounding zone wedge has been breached. A large seafloor feature is located in the eastern side of the grounding zone wedge. This feature resembles a mass wasting deposit, or a slope failure, generated from the grounding zone wedge. However, this feature could also be the result of meltwater
deposition generated when the ice was grounded nearby. Higher data resolution is needed to better characterize this feature and possibly sediment analysis to interpret its depositional origin. Several flutings are present in the seafloor and can be traced from the middle of the cove to the modern glacier front. The flutings are asymmetrical, not parallel to each other, with varying height (5-20 m), width (40-100 m), and length (160-700 m). Unlike flutings in Lever Glacier Cove, the subsurface of these flutings resembles amalgamation or stacking of sediment packages (Fig. 3b), most likely till which has recently been





reworked. A few channels are observed originating near the ice front and flowing towards the middle of the fjord. Channel depths vary between 6-20 m, widths between 30-200 m, and lengths between 70-1200 m. Some of these channels flow between the flutings. The channel axes are easily traced since they are the deepest areas near the ice front.

### 4.4.3 Cadman Glacier Cove

(65°36.7'S, 63°48.7'W) (Fig. 7c) This cove is the smallest in Beascochea Bay, it is 3 km long and 3 km wide, the bay area is about 9 km$^2$. One large glacier drains directly into this cove, Cadman Glacier, with a surface area of 307 km$^2$.

The area surveyed in the cove is insufficient to map large seafloor features, if there are any present in the seafloor. Nevertheless, we identify two large promontories, one on each side of the cove mouth, located in the area that separates the cove from inner Beascochea Bay. The depths along the promontories vary between 180-310 m. These promontories may have been part of a grounding zone wedge at the cove mouth, when Cadman Glacier was grounded at that position. Several slope failures are present on the sides of the promontories. The middle of the cove has a flat basin with water depths down to 550 m.

## 5 Discussion

### 5.1 Distribution and interpretation of seafloor features

We have described numerous seafloor features in four bays in the western AP (additional bays are shown in supplementary material). Many of the bays show similar landform distributions and therefore we propose a schematic model representative of an assemblage of submarine landforms in bays from the western AP (Fig. 8). This continuum of landform features, from the modern ice front to the outer bay area, results from combining the geomorphology of all the bays presented. The inner bay is characterized by flutings and linear channels, and in some cases moraines and crevasse squeeze ridges. The inner bay and middle bay are separated by a grounding zone wedge. The middle bay is characterized by deep, flat-topped basins and, in some cases, drumlins, with their shallow end pointing towards the outer bay. The middle and outer bay are separated by another, likely larger, grounding zone wedge, which is immediately followed by large, asymmetrical, streamlined features. Meltwater channels and smaller, flat-topped basins are common in the outer bay.

Models showing geomorphic features have been presented largely for the continental shelf in Antarctica (Wellner et al., 2001; Canals et al., 2002; Evans et al., 2004; Dowdeswell et al., 2008; Graham et al., 2009), and therefore our model differs from them since we show landforms focused on the confined bay areas. Our proposed schematic model is similar to other models of landform assemblages presented for glaciated environments in Svalbard (Ottesen and Dowdeswell, 2009) and Greenland (Dowdeswell et al., 2016). The main difference is that our model shows more details within the bays and we do not extend our study to the continental shelf as the models from the northern hemisphere. In addition, unlike the bays in





those models, we show that some AP bays include: 1) multiple transverse features, ex. GZW and moraines, 2) meltwater channels throughout the bays, and 3) a transition zone between bedrock and sediment substrate, which results in the formation of different seafloor landforms. Figure 9 shows the actual distribution of submarine landforms per bay compared to the area of the bay and the combined catchment area of the glaciers draining into each bay (also shown in supplementary material). These features were likely formed by ice advancing (and later retreating) throughout the bays and out into the continental shelf during the Last Glacial Maximum. However, features carved in bedrock (e.g. meltwater channels, streamlined features) are likely the result of multiple glaciation cycles in the bays, similar to other areas in the Antarctic continental shelf (Ó Cofaigh et al., 2005; Anderson and Fretwell, 2008; Graham et al., 2009; Livingstone et al., 2013). Drumlins, GZW, moraines and basins appear in lower numbers in the bays, while streamlined features, flutings, slope failures, and meltwater channels are more pervasive (Figs. 9 and 10). The submarine landforms are classified into four categories based on their depositional environment and sedimentary processes forming them: 1) subglacial landforms, 2) ice-marginal landforms, 3) landforms formed by glacial retreat and minor re-advance events, and 4) recent sediment reworking throughout the bays.

### 5.1.1 Subglacial landforms

Subglacial landforms include streamlined features, meltwater channels, flutings (mapped in the middle of bays) and drumlins (Fig. 2). The streamlined features are found in the outer bays, where actively flowing ice carved them in bedrock, most likely over multiple glaciation events (Anderson and Fretwell, 2008; Graham et al., 2009; Livingstone et al., 2013). These features are not symmetrical and in some areas are more elongated closer to the bay mouth and become stubbier away from the bay. Elongation at the bay mouth may indicate faster ice flow at those locations and then later deceleration as the ice reaches a larger, open area to flow outside of the confined bay (Bradwell et al., 2008). Subglacial meltwater channels have been carved into crystalline bedrock, mostly in the outer bay areas. In addition, the number of channels increases in the southern bays. The meltwater channels mapped in this study area form a complex network of flow, with short straight channels and anastomosing channels. Similar meltwater channels have been mapped in other Antarctic regions with crystalline bedrock (Lowe and Anderson, 2002; Anderson and Fretwell, 2008; Livingstone et al., 2013; Nitsche et al., 2013), and likely also formed through multiple glaciation events. The flutings are interpreted to be formed by flowing ice over a thin deformation till layer; most fluting thicknesses are less than 10 m. Flutes are usually parallel to the ice flow direction, and usually form as ice flow encounters an obstacle, large rock or bedrock, in the up-ice end and subglacial sediment flows to the back of the obstacle, forming an elongated ridge in the lee side (Bennett and Glasser, 2009). Drumlins are observed in the middle of bays, recording ice flow direction towards the bay mouth. They are covered by parallel laminated sediment (Fig. 3) and have been interpreted as forming at the transition between crystalline bedrock and sedimentary strata marking the onset of accelerating flow (Wellner et al., 2001, 2006). This acceleration may be the result of converging flow from different directions in Anvord Bay and Beascochea Bay.



### 5.1.2 Ice-marginal landforms

The grounding zone wedges (GZW) are classified as ice marginal landforms. The GZW are large, transverse sedimentary ridges, usually formed at narrow locations in the bay perimeter. GZW are interpreted as depositional features, formed during stillstands periods during a general ice retreat, when sediment is carried to the grounding line through bed deformation and basal melting (Alley et al., 1989; Anderson, 1999; Dowdeswell et al., 2008; Batchelor and Dowdeswell, 2015). Most of the GZW observed in the western AP bays are asymmetric, with a steep slope distal and gentler slope proximal to the ice front. The geometry of these transverse ridges is similar to much larger GZW in the Ross Sea (Halberstadt et al., 2016) and the Weddell Sea (Campo et al., 2017). The size of the GZW has been correlated with the length of ice stability (Alley et al., 2007; Dowdeswell and Vasquez, 2013; Batchelor and Dowdeswell., 2015), a GZW with a large volume implies a period of longer ice stability. In addition, if the amount of sediment flux is known, then we could estimate the duration of ice grounding at that location (Howat and Domack, 2003).

### 5.1.3 Landforms formed by glacial retreat and minor ice re-advance events

Features formed during glacial retreat and minor re-advance events include moraines, flutings (mapped close to the modern ice front), and crevasse squeeze ridges. Moraines are small sedimentary ridges that can be transverse to the bay length or arcuate, forming a lunate shape across the bay length. The moraines are interpreted to form during ice retreat, but unlike the GZW, the duration of the stillstand is much shorter, and possibly more frequent (Ottesen et al., 2005; Batchelor and Dowdeswell, 2015). The moraines likely formed as ice pushed sediment during a minor readvance, in a general retreat phase (Ottesen and Dowdeswell, 2006; Batchelor and Dowdeswell, 2015). The flutings are interpreted to be formed by flowing ice over a deformation till layer (Bennett and Glasser, 2009), although their internal structure may look different in CHIRP; some have a strong surface with no internal reflectors, while others are formed by stacked sediment packages (Fig. 3). These flutings are located proximal to the modern ice front and therefore are interpreted as recently-formed features. Crevasse squeeze ridges are not a common feature in Antarctica, however they have been observed in the Amundsen Sea embayment (Klages et al., 2013). These landforms have been reported in Iceland (Bennett and Glasser, 2009) and Svalbard (Ottesen and Dowdeswell, 2006, 2009) occurring either as symmetrical, low ridges, or as rhombohedral ridges, about 5 m high, found on the ice proximal margin of moraines. In the bays observed in this study, crevasse squeeze ridges were found proximal to the ice front and in water depths shallower than 120 m. These ridges form by squeezing till in crevasses formed at the base of grounded ice, and they indicate ice stagnation followed by a rapid uncoupling from the seafloor (Powell and Domack, 1995; Ottesen and Dowdeswell, 2006; Bennett and Glasser, 2009). The preservation of these features indicates that no further ice front re-advance has occurred over them.

### 5.1.4 Recent sediment reworking throughout the bays

Features formed by recent sediment reworking include slope failures, linear channels, and deep basin fill. The characteristic steep walls of the glacial valleys have slope failures throughout the geometry of the bays, although in some bays the





submarine mass wasting is more abundant than in others (Figs. 9 and 10). Slope failures also occur in large transverse ridges that result in the formation of either a large fan-shaped feature (Figs. 2f, 7c) or a linear, long, deep channel (Fig. 5b, 6b) that continuously erodes the ridge. The linear channels are proglacial and formed in soft sediment, unlike the subglacial channels carved in bedrock. These channels are differentiated from the subglacial meltwater channels by their linear flow, although

5   some are slightly sinuous, they do not form complex flow networks, some are observed in low numbers or even isolated in a bay (Fig. 6b). These channels may be formed by dense underflows of proglacial meltwater mixed with coarse sediments or by slope failures in unstable areas (Syvitsky et al., 1987; Dowdeswell and Vasquez, 2013). The proglacial channels carry sediment flows from bathymetric high regions to deep basins, where sediment of varying sizes is deposited in layers (Fig. 3), forming the laminated basin fill (Powell and Domack, 1995; Ó Cofaigh and Dowdeswell, 2001; Domack et al., 2006).

10  Although, some of the basin fill sediments could have also been deposited subglacially through meltwater flow (Domack et al., 2006), through tidal pumping, close to the grounding line (Domack, 1990), or by meltwater plumes originating at the glacier terminus (Domack et al., 1994; Domack et al., 2006). Sediment reworking processes have likely been occurring since grounded ice started retreating, however, the recent warming period in the AP area may have contributed to an increase in meltwater production which may have resulted in larger sediment reworking.

### 5.2 Observations on ice flow dynamics

There is a variable spatial distribution of the submarine landforms presented in this study (Fig. 1). Although, we present a generic model representative of the geomorphology in the western AP bays (Fig. 8), it is clear that not all the features are present in all the bays and therefore we examined the local conditions in order to understand ice flow in the western AP.

### 5.2.1 Latitude and temperature gradient

Although there are some latitudinal differences between the bays observed, Marian Cove (62°12'S) at the north, and Cadman Glacier Cove (65°36'S) at the south, we did not find a direct correlation between latitude and the number of features found in the bays (Fig. 11a). However, the number of flutings and meltwater channels increased towards the southern AP (Figs. 9 and

25  10). In addition, the complexity of the meltwater channel flow networks increased towards the south AP. This is consistent with the higher latitudinal locations of meltwater channels identified in other studies (Lowe and Anderson, 2002; Anderson and Fretwell, 2008; Livingstone et al., 2013; Nitsche et al., 2013).

In northern locations in our study area, the bay seafloor has a lower relief, e.g. Maxwell Bay (Fig. 4) and Hope Bay (Fig. 5).

30  Although there are some bathymetric highs, the seafloor appears smoother overall. In comparison, the southern bays have a more rugged seafloor, with very high differences in relief, e.g. Flandres Bay (supplementary material Fig. 5) and Beascochea Bay (Fig. 7). The deep basins with flat tops in Flandres Bay and Beascochea Bay contrast with the variable relief around them. We attribute these differences in seafloor roughness to a higher sediment cover in the northern areas compared to the southern areas, and not necessarily to differences in bedrock topography. The increased sediment cover is related to higher




sediment accumulation rates, documented in Maxwell Bay by Milliken et al. (2009) and Boldt et al. (2013). Thus the lack of geomorphic features on the modern seafloor cannot be interpreted to indicate lack of their formation. In fact, the smooth seafloor cover in Maxwell Bay has more resemblance with Chilean fjords (e.g. Dowdeswell and Vasquez, 2013) and bays in South Georgia (e.g. Hodgson et al., 2014) than it does with other AP bays.

### 5.2.2 Bay area and glacier drainage area

One of the apparent variables in this comparison is the size of the bay area and the catchment area of the glaciers draining into any particular bay (Fig. 9). A reasonable assumption is that a larger drainage area would likely result in larger amounts of sediment delivered to the seafloor, which could potentially form more seafloor features as ice flows in the bay. In

addition, a larger bay would likely have a larger number of landforms. We compared bay area and glacier catchment area (total combined area of the glaciers draining into each bay) to the number of features mapped in the bays. We found a relatively high correlation between the bay area and the number of features (Fig. 11e), and a very poor correlation with total glacier catchment area (Fig. 11f). Larger bays have, in average, more submarine landforms, but a larger drainage area does not result in more submarine landforms in the bay. This conclusion implies that landform formation is complex and not

directly dependent on the amount of ice flow into the bays.

When comparing glacier catchment areas to the type of features found (Fig. 9), the smaller drainage areas are correlated with the smaller size features, ex. moraines and crevasse squeeze ridges, which are not found in bays with larger catchment drainages. This suggests that smaller fluctuations in the ice flow (that would result in the formation of smaller landforms)

would not be apparent in larger glacial systems. Therefore, we conclude that the size of the bays, and not the size of the catchment area, dictates the number of features that form in the seafloor, but smaller glacier catchment areas are able to record evidence of small fluctuations in ice flow. This conclusion is consistent with results from Bourgeois Fjord and Blind Bay (Garcia et al., 2016), near Marguerite Bay in the southern AP, where an inverse relation between drainage basin size and retreat of the glacier terminus was found. Similarly, Fox and Cooper (1998) measured the largest size reduction on the

smaller ice bodies in the AP.

### 5.2.3 Geometry of bays

Since there is a large degree of variability regarding size of the bays (Fig. 9), we additionally analysed the bay length, bay width, and bay ratio (length/width). Bays with ratios lower than 1 were classified as open bays, ratios between 1 and 2 were

30 classified as broad bays, and ratios higher than 2 were classified as narrow bays. We refer to this classification as the geometry of the bays. This geometry was compared to the type (and number) of features found in each observed bay (Fig. 10 and Table 1 in supplementary material). Because most bays in our study area were classified as narrow, we use "percentage of bays" as a way to normalize the results. Therefore, we refer to the percentage of narrow (or combined broad and open)



bays where certain feature was identified; for example, Fig. 10(b) shows that 100% of the narrow bays have GZW, while only 42% of the broad/open bays have this same type of feature.

In Fig. 10(b) we note that crevasse squeeze ridges, drumlins, and moraines occur only in narrow bays; GZW occur mostly in narrow bays; and streamlined features, slope failures, and meltwater channels occur mostly in broad/open bays. Basins and flutings tend to occur in all bays regardless of the bay geometry, and no feature type occurs exclusively in open bays. In addition, we compared bay length, bay width, and bay ratio to the number of features mapped (Fig. 11 b-d). We see that both, bay length and width, have a strong correlation with the number of features found in the bay, while bay ratio does not show any correlation.

From these observations we conclude that bay length and bay width exert a large control over the number of features found, and that the geometry of the bay dictates the types of features that form. Narrower bays tend to form transverse features, like moraines and GZW, which form during periods of ice stabilization (Anderson, 1999; Alley et al., 2007; Dowdeswell et al., 2008; Dowdeswell and Vasquez, 2013; Batchelor and Dowdeswell., 2015). The width of the glacial valley has been suggested to play an important role for glacial flow (O'Neel et al., 2005; Joughin et al., 2008; Robel et al., 2017). Similarly, widths of ice-stream troughs, along with water depth, control ice flow by increasing the lateral resistance (Whillans and van deer Veen, 1997; Jamieson et al., 2012). The lateral drag increases as the width narrows which may lead to ice stabilization that could result in transverse features, based on the amount of sediment flux and duration of the still-stand (Howat and Domack, 2003; Dowdeswell and Vasquez, 2013). Transverse-to-flow features in the Ross Sea and Weddell Sea (Halberstadt et al., 2016; Campo et al., 2017) are larger than the GZW and moraines identified in this study and are the result of a much larger ice flow system. Therefore, width may play a major role in confined flow, e.g. fjords and bays.

### 5.2.4 Seafloor lithology

The CHIRP dataset was used to identify the lithology and sediment thicknesses of some of the features mapped in the bays (Fig. 3). Five facies were identified throughout the study area and two localized examples (drumlins and flutings) because they present a unique internal acoustic configuration. The GZW and moraines are characterized by a strong surface with no internal reflectors; basins have thinly laminated sediments; the streamlined features and the meltwater channels were both carved in bedrock and they form bowtie or hummocky reflections; drumlins have a thin sediment cover; and flutings show either a hard surface and no internal reflectors or they show thick, amalgamated packages of reworked sediment.

Two different types of channels were identified in this study: 1) linear, wide channels, carved in soft sediment, interpreted as a modern erosional feature. These types of channels are common in Chilean bays (Dowdeswell and Vasquez, 2013) and northern hemisphere fjords (Syvitski et al., 1987; Bennett and Glasser, 2009) where they form by dense sediment flows or turbidity currents resulting from glacifluvial meltwater or slope failures. And 2) A complex network of channels carved in



bedrock, interpreted as subglacial meltwater channels, that highlight the production of subglacial meltwater in the northern AP region, previously only identified in southern AP areas (Dowdeswell et al., 2004; Anderson and Fretwell, 2008; Livingstone et al., 2013).

5 Drumlins were found in only two occurrences, both on very large bays (Fig. 9). In both examples, the drumlins occur in the middle bay area, at a convergence of ice flow from two large drainage systems. This convergence may result in flow acceleration, which would explain the formation of the drumlins, which are then followed by flutings (Wellner et al., 2001; 2006).

## 5.2.5 Water depth

Three of the bays analysed have shallow water depths, less than 150 m deep; Marian Cove, Potter Cove, and Hope Bay. All these bays are characterized by a series of retreat moraines throughout the length of the bay (Figs. 4 and 5). In addition, in two of these bays (Potter Cove and Hope Bay) crevasse squeeze ridges are present in the ice proximal area (Figs. 4, 5, and 9). The presence of the moraines indicates short periods of ice advance, during a general ice retreat (Ottesen et al., 2005; Ottesen and Dowdeswell, 2006; Batchelor and Dowdeswell, 2015), while the crevasse squeeze ridges indicate a period of ice stagnation followed by ice decoupling from the seafloor (Ottesen and Dowdeswell, 2006; Bennett and Glasser, 2009). These same set of features, moraines and crevasse squeeze ridges, are more common in Svalbard (Ottesen and Dowdeswell, 2006; Ottesen et al., 2008; Ottesen and Dowdeswell, 2009) where water depths are shallower than 150 m. In this study, shallow water depth is correlated with features that suggest rapid lift-off of grounded ice from the seafloor and subsequent retreat.

Therefore, shallower water in the bays could lead to rapid ice decoupling from the seafloor, followed by ice front retreat, as indicated by the formation of moraines and crevasse squeeze ridges.

## 5.3 Comparison to other glaciated regions

Similar assemblages of submarine landforms are found in bays of Greenland (Dowdeswell et al., 2016) and, to a lesser extent, in Svalbard (Ottesen and Dowdeswell, 2009). In Greenland, Dowdeswell et al. (2016) observed lineations near the modern ice front followed by a Little Ice Age moraine with channels flowing towards a deep basin in the middle of the fjord, and streamlined features in the outer fjord areas. In Svalbard, several transverse retreat moraines and a larger Little Ice Age moraine ridge characterize the inner bay, followed by drumlinoid features in the middle to outer bay and larger transverse ridges in the outer fjord (Ottesen and Dowdeswell, 2009). Because Svalbard experiences higher sedimentation rates, compared to the AP, it is possible that some of the features seemly not present may actually be covered.

Bays in South Georgia, an island northeast of the AP, also show some similarities; a shallower inner bay, followed by a moraine and a deep basin towards the outer bay (Hodgson et al., 2014). However, many of the bays in South Georgia have smooth seafloors, which indicates any other older features (if any) are likely buried. Dowdeswell and Vasquez (2013)



mapped the geomorphology of some bays near the Southern Patagonian Ice Cap in Chile, and they show less similarities to western AP bays in general. Bays in Chile are dominated by meltwater production that is reworking and redistributing the sediment, draping the seafloor, creating a smooth cover throughout (Dowdeswell and Vasquez, 2013). Much less meltwater production and sediment reworking, along with relatively less sediment cover in the western AP bays, allows to map

submarine landforms in detail. The presence of meltwater channels in the middle to outer bays, carving into bedrock, is an important characteristic that is only present in AP bays and not in the other regions mentioned above.

**5.4 Late Holocene glacial advance**

The seafloor in the ice proximal area in several of the bays presented in this study (Marian Cove, Hope Bay, Lapeyrère Bay,

Fournier Bay, Moser Glacier Cove, Briand Fjord, Lester Glacier Cove, and Funk Glacier Cove) is characterized by a grounding zone wedge (GZW) located a few kilometres from the modern ice front with flutings between the GZW and the modern ice front (Fig. 12). We propose these proximal features are associated with a Little Ice Age (LIA) glacial advance. Similar sets of features (a large, ice proximal transverse ridge followed by either smaller transverse ridges or elongated ridges parallel to the modern ice front) in the inner bays have also been observed in Chile (Dowdeswell and Vasquez, 2013),

Greenland (Dowdeswell et al., 2016), and Svalbard (Ottesen et al., 2005; Ottesen and Dowdeswell, 2009) and have been interpreted as LIA landforms. In Antarctica, much less research has been published associating geomorphology and the LIA. Christ et al. (2014) also observed these same set of features in the ice proximal region of Barilari Bay (Fig. 12j) and referred to them as a "fluted grounding zone wedge". Although, they do not interpret the submarine landforms or the possible formation mechanisms, they do argue for LIA formation based on sedimentological analysis and [210]Pb and radiocarbon

dates. Garcia et al. (2016) describe the geomorphology of a western AP fjord near Marguerite Bay, south of our study area. They show transverse, crescent-shape, and longitudinal ridges ("morainic" landforms), along with elongated ridges, semi parallel to the fjord length in the inner bay. Although they do not present any sedimentological analysis or dating, they interpret these inner features as a result of LIA glacial advance in this fjord because these submarine landforms appear pristine, relatively recent, and are located only a few kilometres from the modern ice front. In Potter Cove, transverse

moraines in the inner cove are likely associated with LIA advance (Wolfl et al., 2016), however no dating was conducted on those features. In the neighbouring Maxwell Bay, no sedimentological evidence was found of LIA advance (Milliken et al., 2009), which may indicate that if there was any LIA advance in the western AP bays, only smaller systems (narrow bays) and/or shallow bays would record and preserve any geomorphic evidence. The LIA event has been reported in western AP bays and the South Shetland Island by several authors (Domack et al., 1995; Shevenell et al., 1996; Domack et al., 2001;

Domack et al., 2003; Hall, 2007; Hass et al., 2010; Monien et al., 2011; Simms et al., 2012) and it may be a more widespread event throughout the AP than previously assumed. However, it is worth noting that LIA interpretations by those authors were based on sedimentological or terrestrial analysis that included results from dating techniques. Our interpretations are based only on geomorphology and, evidently, chronology assessments are necessary to support this argument.



## 6 Summary and Conclusions

We present multibeam swath bathymetry from bays in the South Shetland Islands and the western Antarctic Peninsula. The subglacial landforms were classified into four categories based on their depositional environment and sedimentary processes forming them: subglacial, ice-marginal, glacial retreat and minor re-advance events, and recent sediment reworking. We propose a schematic model showing geomorphic features present in western AP bays; from flutings and moraines in the inner bay, to grounding zone wedges and drumlins in the middle bay, and streamlined features and meltwater channels in the outer bay areas.

We analysed the local variables of each bay including latitude, bay area, bay length, bay width, glacier catchment area, and the seafloor lithology to understand controls on ice flow behaviour. Specific results include the following:

1) Bay length and width exert a control on the number of landform features found in the bays, in addition, the geometry of the bays dictates the types of features that will form. Narrower bays tend to form transverse-to-flow features because the lateral drag of the ice flow increases as the valley width narrows which may lead to ice flow stabilization.

2) The shallowest bays analysed show signs of rapid glacial decoupling from the seafloor and subsequent retreat, which emphasizes the role of water depth as a control on ice flow.

3) Small size features, ex. moraines and crevasse squeeze ridges, were only found in narrow bays with smaller drainage areas, and not in larger-sized drainages areas, suggesting that short-lived environmental fluctuations, responsible for the formation of these features, would only be recorded by the smaller glacial systems.

4) Two different types of meltwater channels were identified: linear, wide channels carved in soft sediment are a modern erosional feature, while the complex network of channels carved in bedrock are subglacial, which highlights the presence of subglacial meltwater production in the northern AP region, possibly through several glacial cycles.

Recently, Cook et al. (2016) concluded that glaciers in contact with warmer ocean water have retreated significantly in the AP region. The presence of deep channels in the outer bay areas may act as conduits to facilitate intrusion of deep, warm water into the inner bays causing basal melting of the tidewater glaciers. Finally, based on analogous assemblages of features reported in other locations, we propose the geomorphic features found in the seafloor of some of the inner bay areas were formed during the Little Ice Age glacial advance. If this is the case, then glacier systems in the AP have a greater sensitivity to minor atmospheric and oceanic fluctuations than previously suggested. Future research should include more multibeam coverage as well as sedimentological analysis and chronometric constrains in order confirm LIA in these bays and in other areas of the Antarctic Peninsula.





**Acknowledgments**

This research was funded by the National Science Foundation, Office of Polar Programs grant no. OPP0739596. YPM was supported by a National Science Foundation Graduate Research Fellowship. We thank the crew and science parties of the RV/IB *Nathaniel B. Palmer* and *Araon* cruises to the Antarctic Peninsula.

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





## Figures



Figure 1: Map of the northern Antarctic Peninsula (AP) and South Shetland Islands, red boxes indicate Figures 4-7. Inset shows the location of the AP in Antarctica. 1 Cierva Cove, 2 Brialmont Cove, 3 Fournier Bay, 4 Charlotte Bay, 5 Anvord
5  Bay, 6 Moser Glacier 7 Lester Cove, 8 Briand Fjord, 9 Flandres Bay, 10 Etienne Fjord, 11 Collins Bay. AP map from Polar Geospatial Center, bathymetry and inset from IBCSO (Arndt et al., 2013).





Figure 2: Seafloor landforms found in the bays of the western Antarctic Peninsula: (a) flutings, (b) moraine and crevasse squeeze ridges, (c) streamlined features, (d) drumlins, (e) channels, (f) grounding zone wedge.

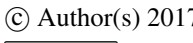


| facies | description | interpretation | chirp example | multibeam example |
|--------|-------------|----------------|---------------|-------------------|
| 1 | Thinly layered | Basin fill | | |
| 2 | Bowtie or hummocky reflections | Bedrock | | |
| 3 | Strong surface, no internal reflectors | Till | | |
| 4 | Thin layer covering a strong reflector | Thin, fine-grained sediment cover | | |
| 5 | Weak reflections, mount shape | Slump, debris flow deposits | | |

**Localized examples**

(a) Asymmetric wedge with reflections: thin sediment cover, drumlin

(b) Stacked sediment packages: channel-fluting pair, near ice front

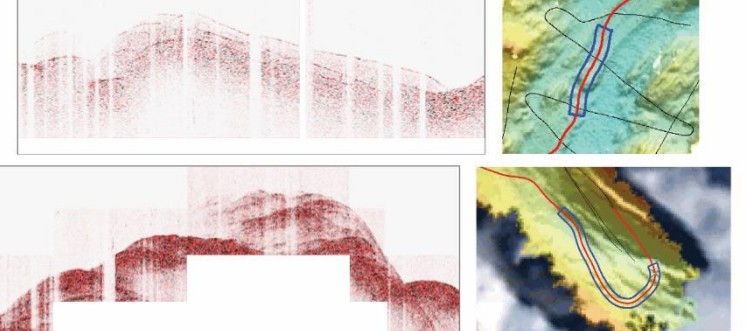

Figure 3: Chirp facies showing seafloor lithology. Five facies were identified throughout the bay, in addition, two localized examples of features are shown: (a) drumlin and (b) flutings near the modern ice front. The blue rectangle in multibeam example shows the location of the chirp example shown.





Figure 4: Multibeam swath bathymetry of Maxwell Bay and surrounding areas, including Potter Cove and Marian Cove, see Fig. 1 for location. Cross sections A-A' and D-D' show transverse moraines in Potter Cove (b) and Marian Cove (c) respectively, B-B' shows elongated, streamlined ridges in the bay, C-C' shows the U-shaped fjord valley. Vertical exaggeration is 5x in all images.



Figure 5: Multibeam swath bathymetry of Hope Bay, see Fig. 1 for location. The inner bay is shown in (b), moraines can be seen in A-A', the U-shaped fjord valley is seen in B-B', and a meltwater channel in the outer bay (C-C'). Vertical exaggeration is 3x in both images.



Figure 6: Multibeam swath bathymetry of Lapeyrère Bay, see Fig. 1 for location. Inner bay is shown in (b). Cross section A-A', B-B', and D-D' show meltwater channels around the bay. C-C' shows the grounding zone wedge in the inner bay. Vertical exaggeration is 3x in both images.





Figure 7: Multibeam swath bathymetry of Beascochea Bay, see Fig. 1 for location. A-A' shows a meltwater channel in the outer bay, (b) and (c) show the inner bay with channels and flutings (C-C'), grounding zone wedges (B-B' and E-E'), and along a drumlin in D-D'. Vertical exaggeration is 3x in (a) and (b), and 5x in (c).



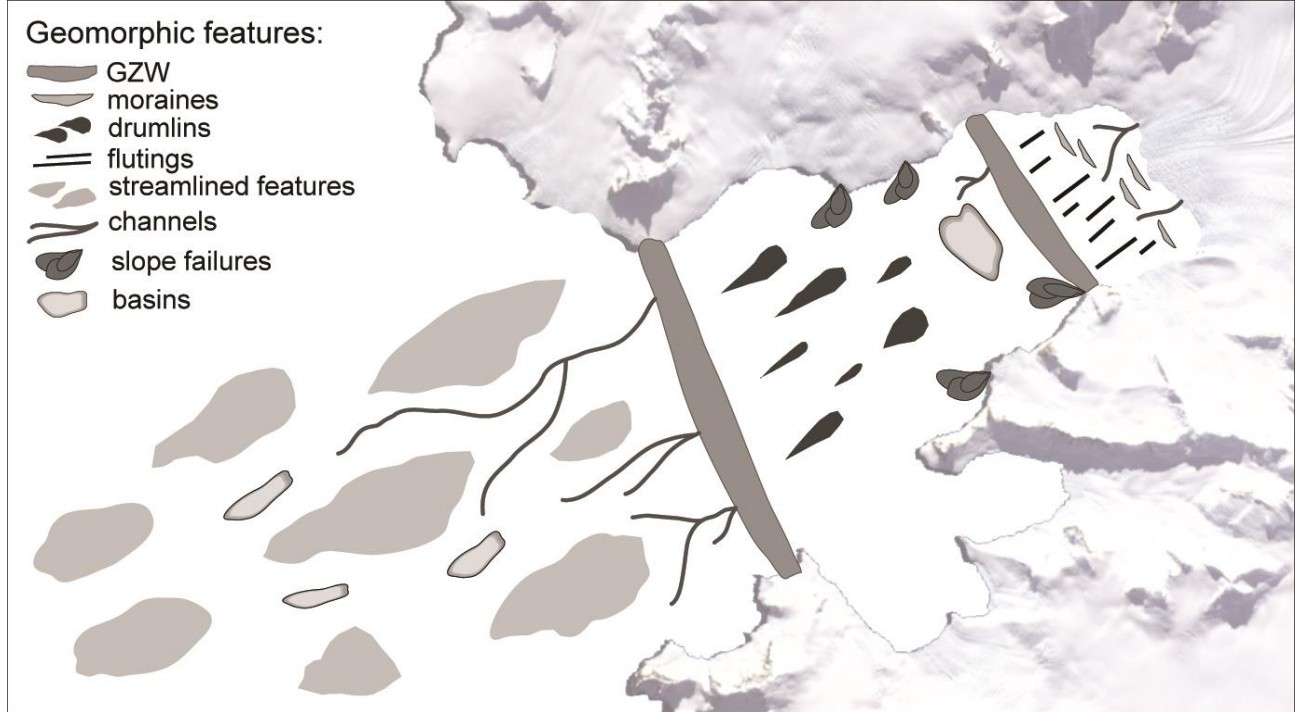

Figure 8: Schematic map view model showing the various geomorphic features found in the seafloor of glaciated bays in the western Antarctic Peninsula.



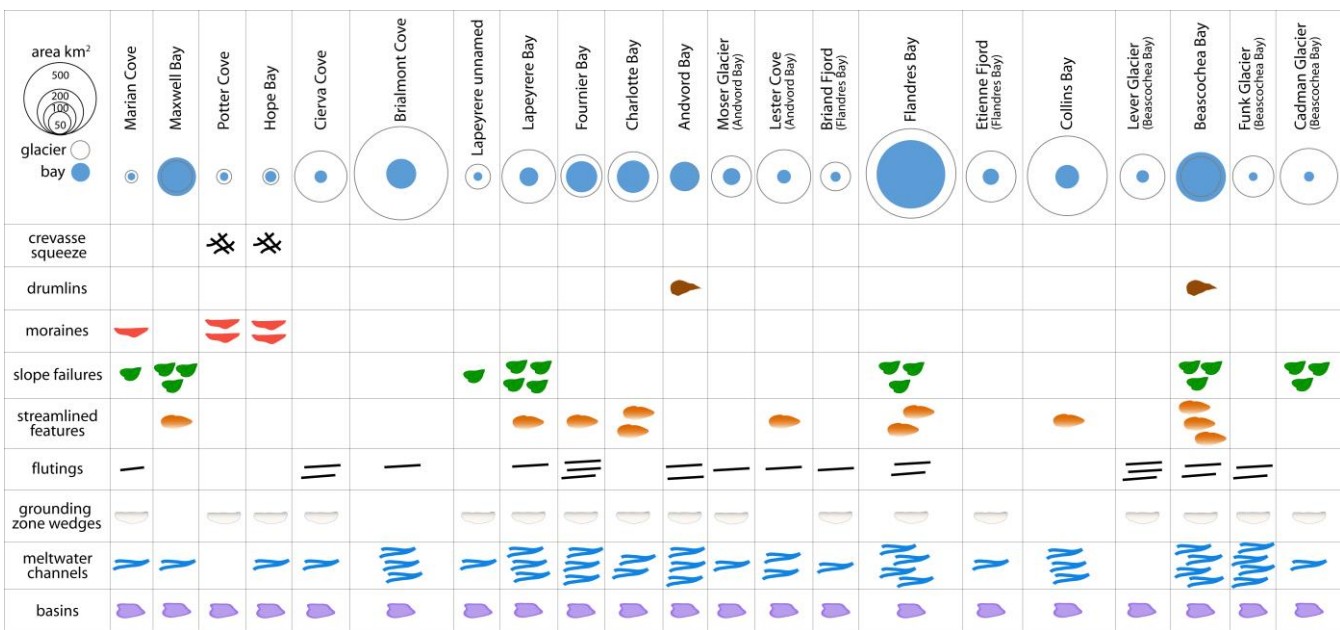

5  Figure 9: Comparing bay area, glacier catchment area, and submarine features found in bays. Bays are listed from northernmost (Marian Cove) to southernmost (Cadman Glacier Cove). The number of symbols in the chart is a representation of the number of geomorphic features found in each bay: one symbol represents less than 10 features, two symbols represents between 10 and 20, three symbols represents between 20 and 30, and four symbols represents greater than 30 features found at that location.



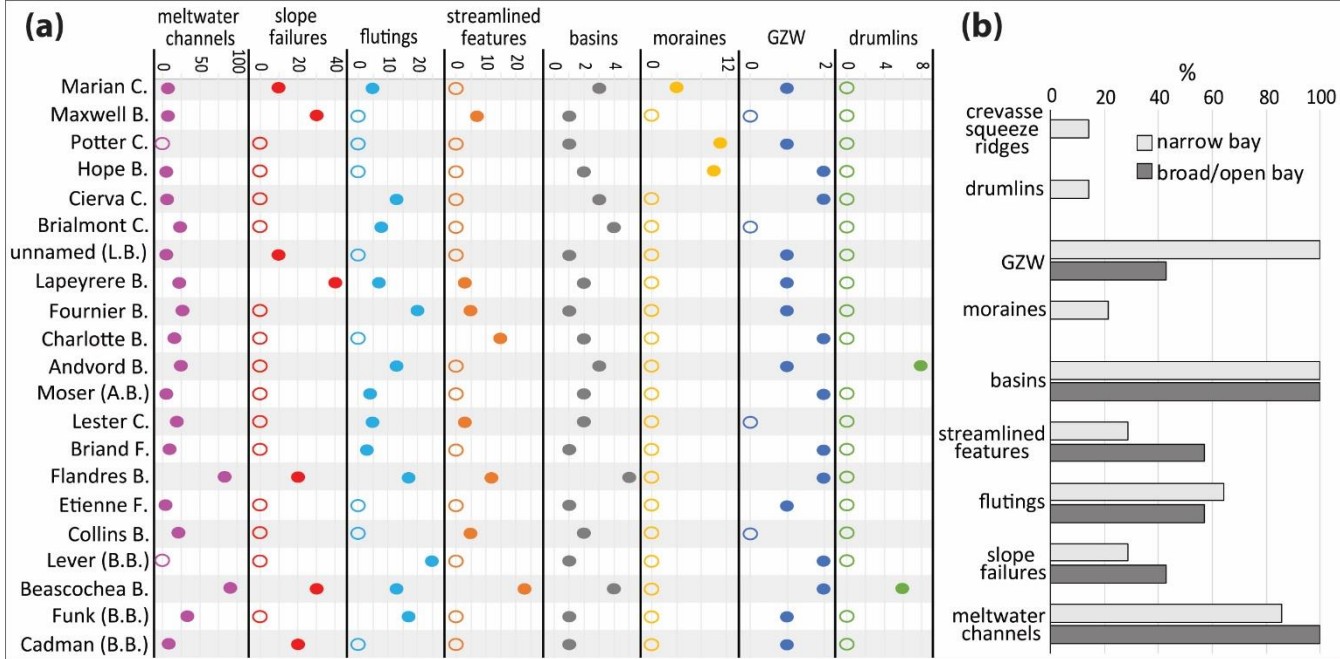

Figure 10: (a) Number of features found in the bays, open circles represent zero; the bays are listed from northernmost to southernmost latitude; C. cove, B. bay, F. fjord, L.B. Lapeyrère Bay, A.B. Andvord Bay, B.B. Beascochea Bay. (b) Percent of narrow (light grey) and broad/open bays (dark grey) in which the listed geomorphic features are found.





Figure 11: Graphs showing number of overall features found in the study areas as they relate to latitude, bay ratio, bay length, bay width, bay area, and glacier catchment area.

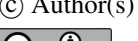

Figure 12: Map of central Antarctic Peninsula (a) and features in the inner bays (Figs. b-j). Figures (b) Marian Cove and (c) Hope Bay are not in map (a), for location refer to Fig.1. Barilari Bay (j) from Christ et al. (2014) shows the proximal grounding zone wedge and flutings, the red square shows the location of sediment core collected in 2010 and used for radiocarbon dating in Christ et al., (2014).