# Peer review of "Seafloor geomorphology of western Antarctic Peninsula bays: a signature of ice flow behaviour"

_The Cryosphere, 2017_

## Referee Comment (RC1) · A. G. C. Graham (Referee) · 2 Aug 2017

In general, the paper by Munoz and Wellner is good. It reports new sea-floor observations from the bays of the AP and makes a sound attempt at analysing the differences between the various sites in terms of geomorphology and relationship to glacier behaviour.

The datasets presented are original, and the results will add nicely to the inventory of shelf and coastal locations already studied along the West Antarctic margin.

The purpose of the work is mostly clear, but the authors might want to consider their

method and whether it is still adequate to simply present sea-floor data without a clear and objective mapping of the landforms alongside. I feel the science has moved on from simply making observations from multibeam data to a more rigorous landform mapping based approach, underpinned by clear metrics. As a minimum, I expected a clearer separation of the descriptions and interpretations of the landforms in the results section, and I feel this is an area where the paper still needs a little more work. Alongside that, some of the interpretations themselves need some better explanation supported by more in-depth analysis and relevant literature. See comments on the attached PDF, but the crevasse-squeeze ridges are a prime example. Some of the multibeam observations might also be better supported by a closer integration and more widespread study of sub-bottom profiler data that is presented as a scheme early on in Fig 3 but referred to only a handful of times before the discussion. Why take an acoustic facies approach if you don't then use it to produce a series of maps?

The discussion is fair/good. There is a bit of repetition which could be lost, and I suggest a few areas where the authors could expand. I remain a little unconvinced by the correlation of landform number to bay size. It seems logical to me that smaller fjords will contain fewer landforms than larger ones. But I am open to persuasion in the authors' response as to whether this constitutes a significant finding or not.

I cannot fault the general attempt to try and correlate the geomorphology to catchment size and geometry, and there are some useful observations borne out by the work. The comparisons to other locations are a little cursory and I would like a more definitive assessment of how AP bays are unique (or similar) to assemblages found in comparable settings around the globe. The LIA discussion is interesting but I felt under-developed. Is it feasible to form a fjord GZW of the sizes you are observing during and since the LIA, based on what you know about sediment fluxes? Why are these all LIA age when Fig 12i clearly shows one of the wedges at least to lie coincidental with a mid-20th century glacier front position?

The paper is likely to make some impact on the community. It spurs on research into

the timing of deglaciation through these various systems, and underpins wider habitat and ecology work in the fjords and bays. To that end, landform and substrate maps would really be useful as an additional product of this work.

Overall, I suggest some moderate revisions based on these comments and those included in the annotated PDF. However, I think all are achievable and the paper will sit well in The Cryosphere with a bit of further refinement.

Please also note the supplement to this comment:
https://www.the-cryosphere-discuss.net/tc-2017-108/tc-2017-108-RC1-supplement.pdf

[Figure]

**Supplement:**

[revised manuscript text omitted]

---

## Referee Comment (RC2) · S.J. Livingstone (Referee) · 23 Aug 2017

This is an interesting, well written paper that compiles glacial geomorphological evidence within a large number of bays across the western Antarctic Peninsula and South Shetland Islands. The information is used to investigate former ice flow behaviour and to infer a Little Ice Age response in the region. The results are novel and could make a nice contribution to TC, but requires major revisions. In particular, I found some of the discussion to be rather weak or speculative in its current state. But the data you have is very nice and so I think the focus should be on doing a really good job of mapping and characterising the glacial imprint of these bays and putting them in the context of other

areas. Some of the more speculative discussion could be chopped and more made of, for instance, the spatial pattern of landforms, variations between location, and data you have compiled on moraines and their location.

General Comments:

1. The authors present the work as a mapping study, following on from other recent publications that have started to do this in Antarctica. However, no mapping is presented in the results/figures. Certainly, this paper would be considerably stronger if the authors could present the mapping results of their work as a series of maps. It would make it much easier for the readers to pick out features and for spatial patterns to be identified and justified. If no mapping was carried out then the term should not be used, although I think this then weakens the paper.

2. The methods is rather short. In particular, there is no clear rationale of how the authors identified the glacial features and interpreted them. I suggest they include a table similar to the one in Graham et al. (2009) – Table 2, or Livingstone et al. (2013) – Table 1. Maybe because of the lack of detail on these landforms I was also left unconvinced about some of the interpretations. For example, you use the term grounding zone wedge (GZW) throughout. But what differentiates them from moraines if they are symmetrical (i.e. without the characteristic 'wedge' shape)? I would stick to the term moraine unless there is clear morphological or stratigraphic (from the CHIRP) evidence that they are GZWs. Similarly, some of the fluting you point out seem more of the scale of lineations, while if drumlins are composed of bedrock maybe whaleback is a better term?

3. The data is really nice and I think a lot more could be made of the results. In particular, some sections lack thorough characterisations of the glacial features (e.g. length, width, shape). Some results and interpretations are mixed up. I always find these section are much clearer when the data is presented first and then interpreted, with support from the literature. For example, on page 5, lines 19-26, you interpret transverse ridges as crevasse squeeze ridges without limited further details of what characteristics allow you to make this leap, and no references to similar interpretations. Some of these issues could be resolved by my suggestion for point 2.

4. The methods describe CHIRP data you have, but this is not really presented until section "5.2.4. Seafloor substrate" in the discussion. This really needs to come in the results, as this provides further evidence that will allow you to interpret the landforms. It would also be useful to identify regions with thick sequences of postglacial infill that could have buried landforms, as this must be an important process (as per the Chile comparison).

5. Meltwater channels. In the discussion, you distinguish between proglacial, postglacia and subglacial channels, but it is not clear what evidence you base this on. In the methods can you outline the criteria you use, and then distinguish between types in the results so it is very clear?

6. The section of basin area and catchment area (and length and width too) is weak. It seems a flawed analysis as of course if you look over a larger area you will find more landforms. Similarly, irrespective of catchment area the number of landforms is likely to be the result of the area you look over, and could be strongly influenced by postglacial burial. The conclusions you draw from this are self-evident.

7. The data you have on moraine/GZW positions is nice, but I think more could be made of it. Why not compile data on their size and where they occur? Do they occur at narrowings or in other contexts (e.g. upglacier of confluences)? Can you investigate whether bays with large changes in width/height have more ice-marginal landforms indicating a stepped pattern of retreat vs. consistent widths, where you might expect more consistent retreat and less evidence of moraines/GZWs.

Specific Comments: See attached pdf.

Please also note the supplement to this comment:

https://www.the-cryosphere-discuss.net/tc-2017-108/tc-2017-108-RC2-supplement.pdf

[Figure]

**Supplement:**

[revised manuscript text omitted]

---

## Editor Comment (EC1) · C. R. Stokes (Editor) · 29 Aug 2017

Dear Yuribia,

You will, by now, have received formal notification that the open discussion of your manuscript has closed. I would like to thank the two reviewers for their constructive comments on your manuscript. Both are encouraging and would like to see the research published, but both identify a number of issues that I would also like to see addressed, and which would likely constitute moderate/major revisions. In particular, this would include a much clearer and explicit presentation of the mapped data, a clearer and more objective separation of the results from the interpretations, and a

more in-depth and critical analysis in the discussion. There be a few instances where you feel there is a good reason not to undertake the suggested revisions, but this would require a detailed rebuttal/justification.

If you have any queries, please do not hesitate to get in touch.

Kind regards,

Chris Stokes Editor

---

## Author Comment (AC1) · 4 Nov 2017

[revised manuscript text omitted]

**Supplementary Material**

**Figures**

[Figure]

**Figure 13.** Interpretation of geomorphic features with hillshade as background in western Antarctic Peninsula bays: (a) Fournier Bay, (b) Charlotte Bay, (c) Andvord Bay, (d) Cierva Cove, (e) Flandres Bay, (f) Brialmont Cove, and (g) Collins Bay. Multibeam data shown in figures 2-7.

[Figure]

**Figure 14.** Multibeam bathymetry map of Hughes Bay, (b) Cierva Cove, and (c) Brialmont Cove. Vertical exaggeration is 3x in (a), and 5x in (b) and (c).

[Figure]

**Figure 15.** Multibeam bathymetry map of Fournier Bay. Vertical exaggeration is 5x in (a) and 3x in (b).

[Figure]

**Figure 16.** Multibeam bathymetry map of Charlotte Bay. Vertical exaggeration is 5x in (a) and (b).

[Figure]

**Figure 17.** Multibeam bathymetry map of (a) Andvord Bay, (b) Lester Cove, and (c) Moser Glacier Cove. Vertical exaggeration is 5x in all images.

[Figure]

**Figure 18.** Multibeam bathymetry map of Flandres Bay, Briand Fjord, and Etienne Fjord. Vertical exaggeration is 5x in (a) and (b).

[Figure]

**Figure 19.** Multibeam bathymetry map of Collins Bay. Vertical exaggeration is 3x in (a) and 2x in (b).

Table 1. Measurements of western Antarctic Peninsula bays. Light grey shows narrow bays, dark grey shows broad and open bays.

| Bay Name | Longitude | Latitude | Bay Length (km) | Bay Width (km) | Ratio length/width | Bay Geometry | Number of Glaciers in Total | Total Glacier Catchment Area (km²) | Bay Area (km²) |
|---|---|---|---|---|---|---|---|---|---|
| Maxwell-Marian Cove | -58.769 | -62.213 | 3.8 | 1.2 | 3.2 | narrow | 1 | 14.99 | 4.868 |
| Maxwell Bay | -58.848 | -62.229 | 15 | 10.4 | 1.4 | broad | 4 | 92.14 | 140.666 |
| Maxwell-Potter Cove | -58.687 | -62.232 | 4.3 | 1.6 | 2.7 | narrow | 1 | 20.26 | 7.093 |
| Hope Bay | -57.046 | -63.407 | 5.8 | 2 | 2.9 | narrow | 4 | 27.36 | 11.448 |
| Hughes-Cierva Cove | -60.87 | -64.155 | 5 | 3 | 1.7 | broad | 2 | 256.36 | 14.056 |
| Hughes-Brialmont Cove | -60.987 | -64.281 | 11 | 7.5 | 1.5 | broad | 5 | 842.53 | 85.500 |
| Lapeyrere's unnamed | -63.286 | -64.37 | 4.6 | 1.8 | 2.6 | narrow | 2 | 60.05 | 7.959 |
| Lapeyrere Bay | -63.284 | -64.421 | 11 | 2.6 | 4.2 | narrow | 7 | 271.06 | 32.496 |
| Fournier Bay | -63.178 | -64.546 | 16 | 5.3 | 3.0 | narrow | 12 | 169.27 | 89.906 |
| Charlotte Bay | -61.611 | -64.614 | 15 | 7.5 | 2.0 | narrow | 13 | 238.14 | 103.153 |
| Advord Bay | -62.566 | -64.857 | 15 | 6 | 2.5 | narrow | 8 | 78.83 | 79.785 |
| Andvord-Moser Gl. cove | -62.425 | -64.872 | 9 | 3.5 | 2.6 | narrow | 6 | 161.14 | 26.027 |
| Andvord-Lester Gl. cove | -62.58 | -64.901 | 5 | 3.8 | 1.3 | broad | 2 | 276.33 | 16.997 |
| Flandres-Briand Fjord | -63.021 | -65.035 | 4.2 | 2 | 2.1 | narrow | 4 | 81.26 | 8.931 |
| Flandres Bay | -63.148 | -65.072 | 30 | 16 | 1.9 | broad | 23 | 774.64 | 453.073 |
| Flandres-Etienne Fjord | -63.234 | -65.171 | 8 | 3.2 | 2.5 | narrow | 7 | 244.5 | 24.587 |
| Collins Bay | -64.054 | -65.346 | 5.6 | 9.5 | 0.6 | open | 4 | 614.11 | 52.430 |
| Beascochea-Lever Gl. cove | -63.723 | -65.512 | 5.3 | 2.5 | 2.1 | narrow | 4 | 190.72 | 16.000 |
| Beascochea Bay | -63.87 | -65.515 | 24 | 10 | 2.4 | narrow | 17 | 155.02 | 235.377 |
| Beascochea-Funk Gl. cove | -63.756 | -65.58 | 4 | 2 | 2.0 | narrow | 2 | 157.77 | 7.874 |
| Beascochea-Cadman Gl. cove | -63.812 | -65.612 | 3 | 3 | 1.0 | broad | 1 | 307.04 | 8.915 |

Table 2. Seafloor features mapped in western Antarctic Peninsula bays. Light grey shows narrow bays, dark grey shows broad and open bays.

| Bay name | crevasse squeeze ridges | moraines | GZW | lineations | drumlins | crag-and-tails | streamlined features | meltwater channels | gullies | basins | total features per bay |
|---|---|---|---|---|---|---|---|---|---|---|---|
| Maxwell-Marian cove | 0 | 6 | 0 | 8 | 0 | 0 | 0 | 2 | 0 | 3 | 19 |
| Maxwell Bay | 0 | 0 | 0 | 0 | 0 | 0 | 15 | 6 | 30 | 2 | 53 |
| Maxwell-Potter cove | 14 | 8 | 0 | 0 | 0 | 0 | 0 | 1 | 0 | 1 | 24 |
| Hope Bay | 18 | 12 | 1 | 0 | 0 | 0 | 0 | 4 | 0 | 3 | 38 |
| Hughes-Cierva cove | 0 | 0 | 2 | 12 | 0 | 0 | 0 | 6 | 0 | 1 | 21 |
| Hughes-Brialmont cove | 0 | 0 | 0 | 3 | 0 | 9 | 0 | 20 | 0 | 4 | 36 |
| Lapeyrere's unnamed | 0 | 0 | 1 | 0 | 0 | 0 | 0 | 3 | 10 | 1 | 15 |
| Lapeyrere Bay | 0 | 0 | 1 | 8 | 0 | 0 | 3 | 20 | 60 | 2 | 94 |
| Fournier Bay | 0 | 0 | 1 | 30 | 0 | 0 | 0 | 14 | 0 | 1 | 46 |
| Charlotte Bay | 0 | 0 | 2 | 0 | 0 | 0 | 12 | 22 | 0 | 2 | 38 |
| Andvord Bay | 0 | 0 | 0 | 8 | 4 | 12 | 0 | 25 | 0 | 1 | 50 |
| Andvord-Moser Gl. cove | 0 | 0 | 3 | 4 | 0 | 0 | 0 | 5 | 0 | 1 | 13 |
| Andvord-Lester Gl. cove | 0 | 0 | 0 | 0 | 0 | 5 | 0 | 4 | 0 | 1 | 10 |
| Flandres-Briand fjord | 0 | 0 | 2 | 3 | 0 | 0 | 0 | 1 | 10 | 1 | 17 |
| Flandres Bay | 0 | 0 | 0 | 14 | 0 | 17 | 12 | 42 | 20 | 6 | 111 |
| Flandres-Etienne fjord | 0 | 0 | 1 | 0 | 0 | 0 | 0 | 4 | 10 | 1 | 16 |
| Collins Bay | 0 | 0 | 0 | 0 | 0 | 6 | 0 | 16 | 0 | 3 | 25 |
| Beascochea-Lever Gl. cove | 0 | 0 | 2 | 28 | 0 | 0 | 0 | 0 | 0 | 1 | 31 |
| Beascochea Bay | 0 | 0 | 1 | 4 | 6 | 13 | 26 | 44 | 30 | 8 | 132 |
| Beascochea-Funk Gl. cove | 0 | 0 | 1 | 17 | 0 | 0 | 0 | 30 | 0 | 1 | 49 |
| Beascochea-Cadman Gl. cove | 0 | 0 | 1 | 0 | 0 | 0 | 0 | 0 | 20 | 1 | 22 |
| total features per type: | 32 | 26 | 19 | 139 | 10 | 62 | 68 | 269 | 190 | 45 | 860 |

---

## Author Comment (AC2) · 4 Nov 2017

**Response to interactive comments for:**

**Seafloor geomorphology of western Antarctic Peninsula bays: a signature of ice flow behaviour**

We want to thank the reviewers and the editor for their constructive and insightful comments. We have made revisions to the manuscript based on the comments provided by the referees. The annotated document with track changes is tc-2017-108-AC1-supplement. Below are answers to specific reviewers' comments followed by the revised manuscript with track changes (underlined are the reviewer comments, followed by the author comments in red).

C. Stokes:
Include 1) a much clearer and explicit presentation of the mapped data, 2) a clearer and more objective separation of the results from the interpretations, and 3) a more in-depth and critical analysis in the discussion.

Y. Munoz:
1) We have included interpretation maps for all the bays mentioned in this paper. In addition, we have included a table describing the criteria used for identifying the features mentioned in the study and we have annotated Fig. 6 which shows examples of the seafloor features mapped.
2) The results have been clearly separated from the interpretations. Results only describe the geomorphology present in the bays as well as geometries of landforms, and interpretation of these landforms is now in the first section of the discussion.
3) The discussion section has been modified to include the interpretation of geomorphic features and some sub-sections have been removed to avoid repetition.

A.G.C. Graham:
1) the authors might want to consider their method and whether it is still adequate to simply present sea-floor data without a clear and objective mapping of the landforms alongside.
We agree, we have included interpretation maps of all the bays presented in this paper.

2) a clearer separation of the descriptions and interpretations of the landforms in the results section
In the revised manuscript, results include only descriptions. Interpretations are listed under discussion.

3) some of the interpretations themselves need some better explanation supported by more in-depth analysis and relevant literature
We have tried to clearly separate features and include more relevant literature for the interpretation.

4) Some of the multibeam observations might also be better supported by a closer integration and more widespread study of sub-bottom profiler data
Unfortunately, we do not have subbottom profiler data paired with the multibeam data. We only have CHIRP data from one cruise, compared to the seven cruises from which we have multibeam data. The CHIRP has been loaded into a project in Kingdom Suite, where we tried applying some filters to clean the data and the results are shown in Figure 7. Given limited areal extent of the data, we did not create lithology maps of the bays.

5) I remain a little unconvinced by the correlation of landform number to bay size. It seems logical to me that smaller fjords will contain fewer landforms than larger ones

We agree, this is intuitive but we still include the correlation because we are also comparing glacier catchment area, and we are suggesting that glacier catchment area does not directly correlate with the number of features regardless of the size of the bay.

6) The LIA discussion is interesting but I felt under-developed. Is it feasible to form a fjord GZW of the sizes you are observing during and since the LIA, based on what you know about sediment fluxes? Why are these all LIA age when Fig 12i clearly shows one of the wedges at least to lie coincidental with a mid-20[th] century glacier front position?

Boldt et al. is one of the few publications that have sedimentation rates for the bays we show in this paper. In several of these bays there are no published sedimentary records and therefore we are inferring deglaciation based on geomorphology alone. Because the proximal area in Barilari Bay (Christ et al.) is very similar to other bays, we assume they may have formed by the same event. But we agree that in some cases, the ice front mapped by Cook et al. coincides with some of the transverse landforms, and these may be younger than LIA. We have included this statement as an alternative idea in the LIA section.

S.J. Livingstone:

1) this paper would be considerably stronger if the authors could present the mapping results of their work as a series of maps

We agree, we now show mapping results for the bays.

2) there is no clear rationale of how the authors identified the glacial features and interpreted them

We have included a table describing the criteria used for identifying the features mentioned in the study and we have annotated Fig. 6 which shows examples of the seafloor features mapped.

3) some sections lack thorough characterisations of the glacial features (e.g. length, width, shape). Some results and interpretations are mixed up.

We have added geometric information of the landforms as detailed as possible in results and the criteria table. We also separated the description (results) from the interpretation of these glacial features.

4) CHIRP needs to come in the results

We have included the CHIRP data in the results, however there is limited coverage and in some areas there are no CHIRP data. The section on seafloor lithology has been removed from the discussion and merged with the results to better characterize the features.

5) Meltwater channels. In the discussion, you distinguish between proglacial, postglacial and subglacial channels, but it is not clear what evidence you base this on

Meltwater channels have been separated into subglacial and proglacial, and in the discussion, we describe the reasoning behind this interpretation.

6) The section of basin area and catchment area (and length and width too) is weak

We agree that the results of this section are intuitive. We still include the correlation because we are also comparing glacier catchment area, and we are suggesting that glacier catchment area does not directly correlate with the number of features regardless of the size of the bay.

7) The data you have on moraine/GZW positions is nice, but I think more could be made of it. Why not compile data on their size and where they occur?

We include size of moraines and GZW in the identification criteria table. We also describe where they occur in the discussion section.

Additional referees' comments in the annotated PDF are not included in this text, but are addressed in the revisions of the manuscript.

The introduction has been modified to explain that this paper is focused in four bays along the Antarctic Peninsula but we integrate data from other 7 locations to support our results. The multibeam maps and the interpretation of features for these 7 locations are included in the supplementary material.

Regional setting has been changed to Study Area to include a short introduction to each one of the four bays and climate in the area.

The discussion section has been modified. We no longer include the category "landforms formed by glacial retreat and minor re-advances", these features have been distributed to either "subglacial", "ice-marginal" or "recent sediment reworking" category. The sections on water depth and seafloor lithology have been removed from "Observations on flow dynamics" to avoid repetition.

[revised manuscript text omitted]
 in western Antarctic Peninsula bays: (a) Fournier Bay, (b) Charlotte Bay, (c) Andvord Bay, (d) Cierva Cove, (e) Flandres Bay, (f) Brialmont Cove, and (g) Collins Bay. Multibeam data shown in figures 2-7.

[Figure]

**Figure 14.** Multibeam bathymetry map of Hughes Bay, (b) Cierva Cove, and (c) Brialmont Cove. Vertical exaggeration is 3x in (a), and 5x in (b) and (c).

[Figure]

Figure 15. Multibeam bathymetry map of Fournier Bay. Vertical exaggeration is 5x in (a) and 3x in (b).

[Figure]

**Figure 16.** Multibeam bathymetry map of Charlotte Bay. Vertical exaggeration is 5x in (a) and (b).

[Figure]

**Figure 17.** Multibeam bathymetry map of (a) Andvord Bay, (b) Lester Cove, and (c) Moser Glacier Cove. Vertical exaggeration is 5x in all images.

[Figure]

**Figure 18.** Multibeam bathymetry map of Flandres Bay, Briand Fjord, and Etienne Fjord. Vertical exaggeration is 5x in (a) and (b).

[Figure]

Figure 19. Multibeam bathymetry map of Collins Bay. Vertical exaggeration is 3x in (a) and 2x in (b).

Table 1. Measurements of western Antarctic Peninsula bays. Light grey shows narrow bays, dark grey shows broad and open bays.

| Bay Name | Longitude | Latitude | Bay Length (km) | Bay Width (km) | Ratio length/width | Bay Geometry | Number of Glaciers in Total | Total Glacier Catchment Area (km²) | Bay Area (km²) |
|---|---|---|---|---|---|---|---|---|---|
| Maxwell-Marian Cove | -58.769 | -62.213 | 3.8 | 1.2 | 3.2 | narrow | 1 | 14.99 | 4.868 |
| Maxwell Bay | -58.848 | -62.229 | 15 | 10.4 | 1.4 | broad | 4 | 92.14 | 140.666 |
| Maxwell-Potter Cove | -58.687 | -62.232 | 4.3 | 1.6 | 2.7 | narrow | 1 | 20.26 | 7.093 |
| Hope Bay | -57.046 | -63.407 | 5.8 | 2 | 2.9 | narrow | 4 | 27.36 | 11.448 |
| Hughes-Cierva Cove | -60.87 | -64.155 | 5 | 3 | 1.7 | broad | 2 | 256.36 | 14.056 |
| Hughes-Brialmont Cove | -60.987 | -64.281 | 11 | 7.5 | 1.5 | broad | 5 | 842.53 | 85.500 |
| Lapeyrere's unnamed | -63.286 | -64.37 | 4.6 | 1.8 | 2.6 | narrow | 2 | 60.05 | 7.959 |
| Lapeyrere Bay | -63.284 | -64.421 | 11 | 2.6 | 4.2 | narrow | 7 | 271.06 | 32.496 |
| Fournier Bay | -63.178 | -64.546 | 16 | 5.3 | 3.0 | narrow | 12 | 169.27 | 89.906 |
| Charlotte Bay | -61.611 | -64.614 | 15 | 7.5 | 2.0 | narrow | 13 | 238.14 | 103.153 |
| Advord Bay | -62.566 | -64.857 | 15 | 6 | 2.5 | narrow | 8 | 78.83 | 79.785 |
| Andvord-Moser Gl. cove | -62.425 | -64.872 | 9 | 3.5 | 2.6 | narrow | 6 | 161.14 | 26.027 |
| Andvord-Lester Gl. cove | -62.58 | -64.901 | 5 | 3.8 | 1.3 | broad | 2 | 276.33 | 16.997 |
| Flandres-Briand Fjord | -63.021 | -65.035 | 4.2 | 2 | 2.1 | narrow | 4 | 81.26 | 8.931 |
| Flandres Bay | -63.148 | -65.072 | 30 | 16 | 1.9 | broad | 23 | 774.64 | 453.073 |
| Flandres-Etienne Fjord | -63.234 | -65.171 | 8 | 3.2 | 2.5 | narrow | 7 | 244.5 | 24.587 |
| Collins Bay | -64.054 | -65.346 | 5.6 | 9.5 | 0.6 | open | 4 | 614.11 | 52.430 |
| Beascochea-Lever Gl. cove | -63.723 | -65.512 | 5.3 | 2.5 | 2.1 | narrow | 4 | 190.72 | 16.000 |
| Beascochea Bay | -63.87 | -65.515 | 24 | 10 | 2.4 | narrow | 17 | 155.02 | 235.377 |
| Beascochea-Funk Gl. cove | -63.756 | -65.58 | 4 | 2 | 2.0 | narrow | 2 | 157.77 | 7.874 |
| Beascochea-Cadman Gl. cove | -63.812 | -65.612 | 3 | 3 | 1.0 | broad | 1 | 307.04 | 8.915 |

Table 2. Seafloor features mapped in western Antarctic Peninsula bays. Light grey shows narrow bays, dark grey shows broad and open bays.

| Bay name | crevasse squeeze ridges | moraines | GZW | lineations | drumlins | crag-and-tails | streamlined features | meltwater channels | gullies | basins | total features per bay |
|---|---|---|---|---|---|---|---|---|---|---|---|
| Maxwell-Marian cove | 0 | 6 | 0 | 8 | 0 | 0 | 0 | 2 | 0 | 3 | 19 |
| Maxwell Bay | 0 | 0 | 0 | 0 | 0 | 0 | 15 | 6 | 30 | 2 | 53 |
| Maxwell-Potter cove | 14 | 8 | 0 | 0 | 0 | 0 | 0 | 1 | 0 | 1 | 24 |
| Hope Bay | 18 | 12 | 1 | 0 | 0 | 0 | 0 | 4 | 0 | 3 | 38 |
| Hughes-Cierva cove | 0 | 0 | 2 | 12 | 0 | 0 | 0 | 6 | 0 | 1 | 21 |
| Hughes-Brialmont cove | 0 | 0 | 0 | 3 | 0 | 9 | 0 | 20 | 0 | 4 | 36 |
| Lapeyrere's unnamed | 0 | 0 | 1 | 0 | 0 | 0 | 0 | 3 | 10 | 1 | 15 |
| Lapeyrere Bay | 0 | 0 | 1 | 8 | 0 | 0 | 3 | 20 | 60 | 2 | 94 |
| Fournier Bay | 0 | 0 | 1 | 30 | 0 | 0 | 0 | 14 | 0 | 1 | 46 |
| Charlotte Bay | 0 | 0 | 2 | 0 | 0 | 0 | 12 | 22 | 0 | 2 | 38 |
| Andvord Bay | 0 | 0 | 0 | 8 | 4 | 12 | 0 | 25 | 0 | 1 | 50 |
| Andvord-Moser Gl. cove | 0 | 0 | 3 | 4 | 0 | 0 | 0 | 5 | 0 | 1 | 13 |
| Andvord-Lester Gl. cove | 0 | 0 | 0 | 0 | 0 | 5 | 0 | 4 | 0 | 1 | 10 |
| Flandres-Briand fjord | 0 | 0 | 2 | 3 | 0 | 0 | 0 | 1 | 10 | 1 | 17 |
| Flandres Bay | 0 | 0 | 0 | 14 | 0 | 17 | 12 | 42 | 20 | 6 | 111 |
| Flandres-Etienne fjord | 0 | 0 | 1 | 0 | 0 | 0 | 0 | 4 | 10 | 1 | 16 |
| Collins Bay | 0 | 0 | 0 | 0 | 0 | 6 | 0 | 16 | 0 | 3 | 25 |
| Beascochea-Lever Gl. cove | 0 | 0 | 2 | 28 | 0 | 0 | 0 | 0 | 0 | 1 | 31 |
| Beascochea Bay | 0 | 0 | 1 | 4 | 6 | 13 | 26 | 44 | 30 | 8 | 132 |
| Beascochea-Funk Gl. cove | 0 | 0 | 1 | 17 | 0 | 0 | 0 | 30 | 0 | 1 | 49 |
| Beascochea-Cadman Gl. cove | 0 | 0 | 1 | 0 | 0 | 0 | 0 | 0 | 20 | 1 | 22 |
| total features per type: | 32 | 26 | 19 | 139 | 10 | 62 | 68 | 269 | 190 | 45 | 860 |